# Self-harm among unaccompanied asylum seekers and refugee minors: protocol for a global systematic review of prevalence, methods and characteristics

Kyli Hedrick [1,2] Rohan Borschmann [2,3]

[1] Department of Public Health and Caring Sciences, Uppsala Universitet, Uppsala, Sweden
[2] Centre for Mental Health, Melbourne School of Population and Global Health, The University of Melbourne, Melbourne, Victoria, Australia
[3] Department of Psychiatry, Warneford Hospital, Oxford University, Oxford, UK

**Correspondence to**
Dr Kyli Hedrick;
kyli.hedrick@unimelb.edu.au

## ABSTRACT

**Introduction** Asylum seekers and refugees are at an elevated risk of self-harm, with younger age and traumatic experiences found to further increase such risk. Despite this, evidence regarding self-harm among unaccompanied asylum seekers and refugee minors has not been synthesised. As self-harm among minors is a risk factor for a range of adverse clinical and social outcomes, including suicide, such information may help to inform evidence-based prevention strategies among these vulnerable populations. This systematic review will synthesise findings from the literature regarding the prevalence, methods and characteristics of self-harm, including risk and protective factors, among unaccompanied asylum seekers and refugee minors internationally.

**Methods and analysis** We searched key electronic databases (PsycINFO, Scopus, PubMed and Ovid MEDLINE) and grey literature for relevant studies published in English from database inception to 10 February 2023. Our primary outcome is self-harm among unaccompanied asylum seekers and/or refugee minors. With the exception of single-case studies, clinical trials and case-control studies, we will include all types of study design that examine the prevalence of self-harm in unaccompanied asylum seekers and/or refugee minors. We will exclude dissertations, conference abstracts, letters, book chapters, editorials, study registrations, registered protocols and qualitative studies. Only studies reporting on participants aged <18 years will be eligible for inclusion. The Methodological Standard for Epidemiological Research Scale will be used to assess the quality of included studies. If there are sufficient studies and homogeneity between them, we will conduct meta-analyses to calculate pooled estimates of self-harm rates, as well as comparisons between subgroups of relevance. If the studies do not report sufficient data, or there is substantial heterogeneity, we will provide a narrative synthesis of the findings.

**Ethics and dissemination** This review is exempt from ethics approval. Our findings will be disseminated through peer-reviewed publications and conference presentations.

**PROSPERO registration number** CRD42021292709.

## STRENGTHS AND LIMITATIONS OF THIS STUDY

⇒ This systematic review uses a comprehensive search strategy including four key academic databases and a grey literature search.
⇒ A strength of this study is that only studies involving unaccompanied asylum seekers and refugee minors and other immigrant populations, and that distinguish between these populations (ie, consider these populations to be heterogeneous), will be included.
⇒ Studies that examine self-harm and suicide attempts but do not distinguish between these acts will be excluded as these two outcomes are qualitatively and motivationally distinct from one another.
⇒ A small number of primary studies, and heterogeneity between study populations and/or study design, may preclude meta-analysis or direct comparison between studies.

## INTRODUCTION

Globally, the number of people forcibly displaced because of war, conflict and political unrest has reached record highs. According to the United Nations High Commissioner for Refugees (UNHCR),[1] as at the end of May 2022, there are now over 100 million forcibly displaced people around the world. Children and young people are markedly over-represented in this humanitarian crisis, accounting for an estimated 41% of those displaced, though they make up 30% of the world's population.[2] Of the 24 100 children who arrived in Greece, Italy, Bulgaria, Spain, Cyprus and Malta in the 12-month period to December 2021, 17 200 (71%) were unaccompanied and separated.[3] The UNHCR defines a refugee as a person who: (1) has a well-founded fear of being persecuted for reasons relating to race, religion, nationality, membership of a particular social group or political opinion; (2) is outside the country of (their) nationality; and (3) is unable, or owing to such fear, is unwilling, to avail (themselves) of the protection of that country (p.14).[4] An asylum seeker is defined as an individual who is seeking refugee 'protection, but whose claim for refugee status has not yet been determined'.[5] Unaccompanied

asylum seekers and refugee minors are defined by the United Nations Convention on the Rights of the Child[6] as children or adolescents younger than 18 years who have been separated from both parents and relatives and are not being cared for by an adult, who, by law or custom, is responsible for doing so.

First-time asylum applications from unaccompanied minors in the European Union (EU) Member States with available data increased by 10% between May and June 2022, and 63% between June 2021 and June 2022.[7] While most unaccompanied minors are males aged between 14 and 17 years, numbers of female and younger populations are increasing.[8] In 2022, the main countries of origin for unaccompanied minors in the EU Member States were Afghanistan, Syria, Somalia and Pakistan.[7] In the 2021 financial year, the USA reported receiving an all-time record of unaccompanied minors from Central America at or near the US–Mexico border, including 145 000 minors.[9]

Many individuals who have been forcibly displaced have been subjected to a range of potentially traumatic events in the pre-migration, peri-migration and post-migration phases.[10] These include witnessing the death of a family member or loved one, war, dangerous flight, poor humanitarian conditions and periods of detention,[10 11] followed by the challenges associated with managing the uncertainty of the asylum process and the resettlement process.[12] Young people who have been forcibly displaced must navigate such adverse experiences during a challenging developmental period, which may further compound the detrimental impact of these events on their mental health.[13] For young people who are forcibly displaced without parents or caregivers, they must not only navigate several potentially traumatic experiences during a difficult time developmentally, but they must do so without the support and guidance of a parent or caregiver, and while managing the loss associated with separation from parents and/or family.[14] Upon arrival in host countries, unaccompanied minors may also be confronted with persistent questioning by authorities about their age, familial relationships and reason(s) for leaving their home country.[15] In addition, they may be subjected to invasive and imprecise age assessments such as carpal, collarbone and/or dental X-rays to establish the accuracy of their self-reported age.[16 17] The particular vulnerabilities and risks faced by unaccompanied asylum seekers and refugee minors have been increasingly acknowledged, paving the way for their entitlement to international protection,[6] as well as programmes promoting their physical, mental and educational development.[6 18]

Research has documented the numerous adverse mental health consequences for unaccompanied asylum seekers and refugee minors across several settings and locations.[19] For example, a 2019 global systematic review of the mental health of unaccompanied refugee minors by von Werthern *et al*[19] found that despite varying estimates of psychological difficulties, rates of post-traumatic stress disorder (17%–85%), depression (13%–76%) and anxiety (11%–85%) were significantly higher in unaccompanied refugee minors than those found in the general western population. Evidence compiled by von Werthern *et al*[19] also found that adolescence and being female were indicators of increased risk of psychiatric disorders. In addition, the authors' review highlighted that there was significantly more exposure to traumatic events such as physical violence, torture, imprisonment and killings among unaccompanied minors, compared with exposure in the general population.[20–23]

Earlier systematic reviews have largely focused on the mental health and well-being of unaccompanied minors,[19 24–26] or the mental health of refugee minors.[27] However, preliminary searches conducted by the authors for the purposes of this study (in PROSPERO, the PubMed/MEDLINE and DARE databases, and the JBI Evidence Synthesis journal[28]) indicate that no reviews have examined self-harm (elsewhere also described as non-suicidal self-injury (NSSI), and defined as the deliberate, self-inflicted destruction of body tissue, without suicidal intent, and for purposes not socially or culturally sanctioned[29]) in unaccompanied asylum seekers and refugee minors globally. As self-harm is common among young people,[30 31] and this risk is further elevated among those with experiences of trauma,[32] it is conceivable that unaccompanied asylum seekers and refugee minors are at even greater risk of self-harm due to their younger age and prior experience of a range of potentially traumatic events in the pre-migration, peri-migration and post-migration phases.[10] Should a higher prevalence of self-harm among unaccompanied asylum seekers and refugee minors be identified, these findings would elucidate the need for providing mental health support at the earliest available opportunity for such minors. Indeed, given that self-harm is a risk factor for severe mental illness, as well as for all-cause mortality,[33] self-harm is an outcome of marked importance. Furthermore, as every act of self-harm (including explicitly non-suicidal actions and behaviour) has the potential to be lethal as a result of accident or misadventure, and self-harm is the strongest risk factor for suicide,[33 34] self-harm (irrespective of intent) is an extremely important outcome to target. Furthermore, as the social and public health costs of self-harm are known to be high,[34 35] the synthesis of such evidence may help to inform distribution of scarce public resources. One broader review that focused on self-harming behaviours among immigrants in Europe by Gargiulo *et al*[36] did report—via a synthesis of three pertinent studies—that unaccompanied minors were an 'at-risk' group for self-harm in the European context. However, as unaccompanied minors were not explicitly included in Gargiulo *et al*'s[36] search strategy, and the setting of their review was European, rather than global, it is likely that other salient studies may have been excluded. As such, a comprehensive global review synthesising the evidence relating to self-harm among

unaccompanied asylum seekers and refugee minors could inform evidence-based prevention and management strategies in these vulnerable populations.

A Swedish study by Ramel et al[37] was included in both von Werthern et al[19] and Gargiulo et al[36] reviews. In this study, the authors compared the psychiatric care of unaccompanied refugee minors and accompanied refugee minors (with an average age of 15.1 years), reporting that 3.4% (or 56 of 1657) of unaccompanied refugee minors received inpatient psychiatric care and 0.67% involuntary care, compared with 0.26% (205 of 81 077) and 0.02%, respectively, of the accompanied refugee minors.[37] Both comparisons were statistically significant (p<0.001). In addition, Ramel et al found that a higher proportion of unaccompanied refugee minors (21.8; 76%) than accompanied minors (21.1; 56%) exhibited self-harm or suicidal behaviour on admission to hospital (p<0.001).[37] The study also noted that 86% (13.2) of unaccompanied minors were hospitalised due to the stress associated with the asylum process.[37] A 2022 systematic review of the mental health of unaccompanied minors in Europe by Daniel-Calveras et al[38] also included one retrospective study evaluating data from 101 unaccompanied refugee minors (aged 14–17 years) presenting to healthcare providers in the UK, reporting that 8% (8 of 101) had engaged in self-harm or suicidal behaviour.[39] None of the studies included in the reviews by von Werthern et al,[19] Gargiulo et al[36] or Daniel-Calveras et al[38] (or the original authors of the two studies highlighted above[37 39]), however, separated out or distinguished—where possible—episodes of self-harm from suicidal behaviour when presenting their findings, though Gargiulo et al[36] did state in their discussion that self-harm should be viewed as a distinct clinical entity. Indeed, definitions of self-harming behaviour have varied historically,[40 41] and behavioural intent is difficult to measure. Despite this, as self-harm and suicide attempts are motivationally and qualitatively distinct from one another,[42] differentiating between the two—where possible—is critical to inform the provision of appropriate physical and mental healthcare. Furthermore, as no terms relating to self-harm were included in von Werthern et al's[19] or Daniel-Calveras et al's[38] search strategies, and those included in Gargiulo et al's[36] search strategy were not exhaustive, additional studies focusing specifically on self-harm (as distinct from suicidal behaviour) may not have been identified in these reviews.

We aim to synthesise the evidence regarding the prevalence, methods and characteristics of self-harm, including risk and protective factors, among unaccompanied asylum seekers and refugee minors internationally.

## METHODS AND ANALYSIS

This protocol is reported in accordance with the Preferred Reporting Items for Systematic Reviews and Meta-Analysis Protocols.[43]

### Patient and public involvement

There was no patient or public involvement in the design of this study.

### Eligibility criteria
#### Participants

We will include studies examining self-harm among unaccompanied asylum seekers and refugee minors—defined as a person aged <18 years who has been separated from both parents and relatives and is not being cared for by an adult, who, by law or custom, is responsible for doing so[6]—in any country. Studies reporting on unaccompanied asylum seekers and/or refugee minors and other immigrant populations, but that do not distinguish between these populations, will be excluded. Studies reporting on suicide or suicide attempts only will be excluded. Where studies refer to suicide and self-harm separately, only findings regarding self-harm will be included. Comparison studies involving accompanied asylum seekers and/or refugee minors, as well and the general/native population in countries of origin, transit and/or resettlement, will also be included, where available.

### Outcome measure

Our primary outcome measure will be self-harm. For the purposes of this review, self-harm, elsewhere also described as NSSI, is defined as the deliberate, self-inflicted destruction of body tissue, without suicidal intent, and for purposes not socially or culturally sanctioned.[29] It may include behaviours such as cutting, burning, biting and scratching skin.[29] Self-harm may be measured by self-report, clinical interview or administrative data (eg, emergency department, hospital or incident reports). An additional outcome measure will be method(s) used to self-harm. Methods of self-harm may also be measured by self-report, clinical interview or administrative data, including the WHO's International Classification of Diseases Tenth Revision codes.[44]

### Study design

We will include published cohort studies (prospective and retrospective) of unaccompanied asylum seekers and refugee minors which report on the prevalence of self-harm. We will also include grey literature (see the Information sources and search strategy section below). We will exclude single-case studies, dissertations, conference abstracts, letters, book chapters, editorials, study registrations, clinical trials, case-control studies, registered protocols and qualitative studies. We will not include previous systematic reviews, as not all included studies may not meet our inclusion criteria. We will, however, identify any peer-reviewed studies related to any dissertations, conference abstracts, study registrations and studies assessed for previous reviews that meet our inclusion criteria but were not identified via our search strategy. Only studies published in English will be included.

| Table 1 | MEDLINE search strategy |
| --- | --- |
| 1. | (non-suicidal self-harm or self-injur* or deliberate self-harm or self-poison* or self-mutilat*).af. |
| 2. | (migration or immigration or unaccompanied asylum seeker or unaccompanied refugee minor or child or adolescent or teenager).af. |
| 3. | 1 and 2 |

## Information sources and search strategy

We searched four key health and medical databases (PsycINFO, Scopus, PubMed and OVID MEDLINE) for relevant literature published in English using key terms relating to self-harm, migration and minors, from database inception until the date of our final search (10 February 2023). The MEDLINE search strategy is outlined in table 1. The full search strategy for each database is outlined in online supplemental appendix 1. The electronic database searches will be supplemented by reviewing the reference lists of eligible articles, as well as searching the websites of relevant non-governmental organisations that have worked with unaccompanied asylum seekers and refugee minors (such as Médecins sans Frontières and Save the Children), for pertinent grey literature.

## Study selection

All studies identified through the database search will be downloaded to EndNote[45] and duplicates removed. The remaining studies will be imported into Covidence[46] for screening. All titles and abstracts will be screened for inclusion by the primary author (KH), with a random 20% also screened by the second author (RB). After 20% of the papers identified in the search strategy have been double-screened, we will reassess our eligibility criteria to ensure that they are relevant to the studies that are identified. The reassessment process will involve a discussion between the reviewers, with any differences or uncertainty regarding study inclusion resolved by a third researcher. The overall inter-rater reliability for the title and abstract screening will be calculated using Cohen's kappa statistic.[47] If reliability is low (<0.40),[47] the authors will review the eligibility criteria, double-code a second random 10% of retrieved studies and recalculate Cohen's kappa statistic.[47] Studies will be coded as either 0=does not meet eligibility criteria, or 1=meets eligibility criteria or the full-text article needs to be screened to confirm eligibility. After title and abstract screening is complete, all remaining full-text articles will be independently screened in duplicate by KH and RB, with any conflicts related to study inclusion resolved through discussion with a third researcher. Where clarification is needed to determine eligibility, we will make a maximum of three attempts to contact the original study author(s), with no more attempts at contact made after 2 months.

## Data extraction

Data extraction will be conducted by KH using a standardised extraction form developed by the researchers and will be checked by a second reviewer (RB). The following data will be extracted from each study: author(s), study year, country of study, study design, setting, sample size, sample characteristics (eg, gender, age, country of origin, country/ies of transit and/or resettlement, and length of separation from parents/caregivers, where possible), reported prevalence of self-harm, method(s) used to self-harm, risk and protective factors associated with self-harm, outcome measure(s) used, characteristics of findings including incidence or episode rates, 95% CIs, p values and effect sizes (where reported). A maximum of three attempts will be made to contact study authors for further information if any of the required data are missing, incomplete or unclear.

## Risk of bias

The Methodological Standard for Epidemiological Research (MASTER)[48] Scale will be used to assess the quality of included studies. The quality and risk of bias will be independently appraised in duplicate by KH and RB. The two reviewers will then compare scores with any uncertainty resolved through discussion and consensus before allocation of final appraisal scores. The MASTER Scale[48] provides a single consolidated tool to assess the risk of bias across different types of study design. This is done by assessing each study for the presence of several methodological standards aimed at addressing the risk of bias across six potential bias domains (selection, information, design related, analytical, confounding and external validity).[48] Using the approach taken in previous systematic reviews,[49 50] we will discuss the possible risk of bias and study quality in text, as well as generate a score summarising each study's quality by using the proportion of safeguards against bias that each study incorporated.

## Data synthesis

We will provide a descriptive overview of the included studies, including the study year(s), design, size and location of the study sample, measure(s) used to report self-harm and any associated characteristics.

If a sufficient number of studies report on the rates of self-harm, or associated characteristics of self-harm, and there is sufficient homogeneity between studies (by, for example, design, sample or outcome), we will conduct meta-analyses to calculate pooled estimates of self-harm rates. Heterogeneity will be assessed using the $I^2$ statistic.

If the number of included studies is sufficient, we will use meta-regression[51] to investigate the influence of gender, age, country of origin and length of separation from parents/caregivers on rates of self-harm. To investigate the impact of study quality on risk of bias, we will conduct a sensitivity analysis which includes only papers rated as high quality (ie, those papers assessed as scoring above the median in the MASTER[48] Scale).

If meta-analyses are not possible due to insufficient data on the rates or associated characteristics of self-harm, or if there is substantial heterogeneity, a narrative synthesis will be created (as per established guidelines on the conduct of a narrative synthesis[52]). This descriptive overview will provide information in both text and table format to summarise and explain the included study findings, based on outcome, associated characteristics of self-harm and geographical region. From this, we may also develop a theory[52] of self-harm among unaccompanied asylum seekers and refugee minors, and the factors that influence it, and an assessment of the overall robustness of this summary.

## Ethics and dissemination

As this is a review of studies that have already obtained ethics approval, this study is exempt from ethics approval. The findings of our review will be disseminated in a peer-reviewed journal article and via presentations at relevant national and international conferences.

**Contributors** KH developed the original research proposal. KH and RB both contributed to the design of the project. KH developed the search strategy, with input from RB. KH wrote the initial draft of the manuscript, and RB contributed significantly to drafting and editing subsequent iterations. Both authors approved the final manuscript.

**Funding** This research received no specific grant from any funding agency in the public, commercial or not-for-profit sectors. RB receives salary and research support from a National Health and Medical Research Council Emerging Leadership Investigator Grant (EL2; GNT2008073).

**Competing interests** None declared.

**Patient and public involvement** Patients and/or the public were not involved in the design, or conduct, or reporting, or dissemination plans of this research.

**Patient consent for publication** Not required.

**Provenance and peer review** Not commissioned; externally peer reviewed.

**ORCID iDs**
Kyli Hedrick http://orcid.org/0000-0002-5546-8116
Rohan Borschmann http://orcid.org/0000-0002-0365-7775

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
