## [Reviewer comments · BMJ Open]

ARTICLE DETAILS

TITLE (PROVISIONAL)	Self-harm among unaccompanied asylum seekers and refugee minors: protocol for a global systematic review of prevalence, methods, and characteristics
AUTHORS	Hedrick, Kyli; Borschmann, Rohan

VERSION 1 – REVIEW

REVIEWER	Uphoff, Eleonora University of York
REVIEW RETURNED	27-Oct-2022

GENERAL COMMENTS	This is a transparent, clearly written protocol on an important topic. Some of the review methodology, for example the lack of double screening, does not adhere to standards for a high quality systematic review. The authors may wish to add an explanation for using a pragmatic review approach. Comments/ suggestions: - The search strategy is described on page 10. It would be better, and easier to read, if this was replaced with a short sentence. For example: "The search included key terms relating to self-harm, migration, and minors." The full search strategy, for one database at least, can then be added as an appendix. Given that the PROSPERO record shows the review was planned to be completed in June 2022, a detailed search strategy should be available.- Do I understand correctly that full-text screening will be done by one person? If possible, a second person should at the very least check a sample of the records.- "three attempts will be made to contact study authors" It may be helpful to add a deadline here, for example, to say no more attempts at contact will be made after two months.- "It will also explore relationships in the data, the development of a theoretical framework (if relevant), and assess the strength of the evidence for the conclusions drawn from the synthesis, as per formal guidelines on the conduct of a narrative synthesis" The sentence above promises a lot of work being done, without providing any detail. Please explain how these tasks will be done. For example, I would not consider no.17 in the PRISMA checklist completed. You may also elaborate on the way the 'descriptive overview' will be structured, for example by outcome or age group or geographic region.
---

REVIEWER	Courtney, Darren Centre for Addiction and Mental Health, Psychiatry
REVIEW RETURNED	07-Jan-2023

GENERAL COMMENTS	The manuscript is a protocol for a systematic review of prevalence rates. The population are minors (under 18 years old) who are seeking asylum or have refugee status. The outcome variable is self-harm. Exposure/comparator groups and timing of outcomes are not specified. The authors anticipate that prevalence rates will be relatively high and propose that this information could help guide investment in treatment. General: The manuscript is, for the most part, well written, clear and concise. PRISMA-P reporting guidelines are followed. Results of the review may be broadly applicable as the samples of interest may be found across the globe. Some revisions could strengthen the manuscript. (1) Reference to an established method of systematic reviews is needed. One option is to consider reference to the JBI Evidence Synthesis Manual from the Joanna Briggs Institute. Munn Z, Moola S, Lisy K, Riitano D, Tufanaru C. Chapter 5: Systematic reviews of prevalence and incidence. In: Aromataris E, Munn Z (Editors). JBI Manual for Evidence Synthesis. JBI, 2020. Available from https://synthesismanual.jbi.global. https://doi.org/10.46658/JBIMES-20-06 Specifically, this manual guides investigators on: (a) determination of the need for a review (Section 1.2, relevant for page 7, line 3 of manuscript) (b) need to conduct review steps (i.e, title/abstract screening, full-text selection, data extraction, quality assessment), either showing adequate inter-rater reliability/agreement prior to a single rater proceeding with the step; or conducting a step in duplicate (section 5, relevant for methods section). Having a second rater checking the work of another rater is too vulnerable to bias. (2) Reference to the inconsistent literature on self-harm nomenclature would also be helpful. For example,  • O'Carroll, P. W., Berman, A. L., Maris, R. W., Moscicki, E. K., Tanney, B. L., & Silverman, M. M. (1996). Beyond the Tower of Babel: a nomenclature for suicidology. Suicide and Life-Threatening Behavior, 26(3), 237-252. • Silverman, M. M., Berman, A. L., Sanddal, N. D., O'carroll, P. W., & Joiner Jr, T. E. (2007a). Rebuilding the tower of Babel: a revised nomenclature for the study of suicide and suicidal behaviors. Part 1: Background, rationale, and methodology. Suicide and Life-Threatening Behavior, 37(3), 248-263. • Silverman, M. M., Berman, A. L., Sanddal, N. D., O'carroll, P. W., & Joiner Jr, T. E. (2007b). Rebuilding the tower of Babel: a revised nomenclature for the study of suicide and suicidal behaviors. Part 2: Suicide-related ideations, communications, and behaviors. Suicide and Life-Threatening Behavior, 37(3), 264-277. • De Leo, D., Goodfellow, B., Silverman, M., Berman, A., Mann, J., Arensman, E., ... & Kolves, K. (2021). International study of definitions of English-language terms for suicidal behaviours: a
--

	survey exploring preferred terminology. BMJ open, 11(2), e043409. Hawton and colleagues' definition of self-harm does include suicide attempts; several times in the manuscript authors suggest that self-harm is mutually exclusive from suicidal behaviour – when this is not the case in Hawton's definition; this needs further clarification. (3) The term “unaccompanied asylum seekers or refugee minors” is confusing. Namely, all of the adjectives refer to minors. I wonder if “unaccompanied minors who are refugees or seeking asylum” might be a better fit; an abbreviated version (e.g, UMRSA) could be used once defined. I also recommend that the operational definitions of these terms can be moved in the earlier part of the introduction as it contextualizes the literature to date described. (4) The potential impact of the results of the current study should be made clearer. If a high prevalence is found, is there a way to determine if it is higher than the general population <18 years old (which is already high)? Is the goal to establish whether refugee status/asylum-seeking is an independent risk factor for self-harm? How could the results specifically inform treatment efforts? Why is self-harm an important outcome to target in treatment, particularly if not suicidal? (5) Prior literature should be further clarified. When a study is referred to, the study design and sample size and age ranges need to be clearly articulated to justify the need for the current study. If proportions are listed, indicate 95% CI's or p-values to say how the samples are different. (6) Further detailed recommendations: (a) In the abstract, the inclusion/exclusion criteria need to match the methods section. In the abstract it seems very broad, in the methods section it seems very narrow: “cohort studies”. (b) EU needs to be spelled out on the first instance. (c) Pg 6, In 36: When it's a group of authors being cited, add “and colleagues...” (d) Pg 6, In 57: “Neither” □ “None” (e) Inclusion/exclusion: What about case-control studies? Qualitative studies? Grey literature? Published protocols? Clinical trials? (f) Specific search strategy can go in an Appendix/Supplementary material, with a brief general description in the main text. (g) Having a priori criteria to progress to a meta-analysis would be ideal (e.g., number of papers, consistency across papers in design/samples/outcome measures, etc.).
--	--

VERSION 1 – AUTHOR RESPONSE

Reviewer: 1

Dr. Eleonora Uphoff, University of York

Comments to the Author:

This is a transparent, clearly written protocol on an important topic.

Thank you for your valuable feedback and time, we are pleased to hear that you believe our protocol is transparent, and clearly written.

Some of the review methodology, for example the lack of double screening, does not adhere to standards for a high-quality systematic review. The authors may wish to add an explanation for using a pragmatic review approach.

In response to your comments, as well as those from the second peer reviewer, we have now clarified in the methods section (pages 10-11), that double screening will occur. In line with further feedback from reviewer 2, we have also added additional details about the screening and rating process (pages 10-11). The full text regarding the screening process (pages 10-11) now reads:

'All studies identified through the database search will be downloaded to Endnote⁴⁵ and duplicates removed. The remaining studies will be imported into Covidence⁴⁶ for screening. All titles and abstracts will be screened for inclusion by the primary author (KH), with 20% screened by the second author (RB). After 20% of the papers identified in the search strategy have been double screened, we will reassess our eligibility criteria to ensure that they are relevant to the studies that are identified. The reassessment process will involve a discussion between the reviewers, with any differences or uncertainty regarding study inclusion resolved by a third researcher. The overall inter-rater reliability for the title and abstract screening will be calculated using Cohen's kappa statistic.⁴⁷ Studies will be coded as either 0=Does not meet eligibility criteria, or 1=Meets eligibility criteria or the full-text article needs to be screened to confirm eligibility. After 20% of the citations have been screened, the eligibility criteria will be reassessed through discussion between all researchers to ensure that they are relevant to the studies that have been identified. If applicable, the updated eligibility criteria will be used for the remaining screening.

After title and abstract screening is complete, all remaining full-text articles will be independently screened by KH and RB, with any conflicts related to study inclusion resolved through discussion with a third researcher. Where clarification is needed to determine eligibility, we will make a maximum of three attempts to contact the original study author(s), with no more attempts at contact made after two months.'

Comments/ suggestions:

- The search strategy is described on page 10. It would be better, and easier to read, if this was replaced with a short sentence. For example: "The search included key terms relating to self-harm, migration, and minors." The full search strategy, for one database at least, can then be added as an appendix. Given that the PROSPERO record shows the review was planned to be completed in June 2022, a detailed search strategy should be available.

Thank you for this feedback. We have now included one full search strategy (MEDLINE) as table 1 (page 10), as well as outlining for the reader that 'the full search strategy used for each database is outlined in Online supplemental appendix 1' (page 10). This is also consistent with feedback from the editor (see above), as well as reviewer 2 (see below). In addition, in response to your suggestion, we have outlined the key search terms related to self-harm, migration, and minors (page 10).

- Do I understand correctly that full-text screening will be done by one person? If possible, a second person should at the very least check a sample of the records.

In response to your feedback, as well as comments from the second reviewer, we have now further clarified the screening process in the methods section (pages 10-11) in order to highlight that full-text screening will be conducted by more than one person. The updated text now reads:

'After title and abstract screening is complete, all remaining full-text articles will be independently screened by KH and RB, with any conflicts related to study inclusion resolved through discussion with a third researcher. Where clarification is needed to determine eligibility, we will make a maximum of three attempts to contact the original study author(s), with no more attempts at contact made after two months.'

- "three attempts will be made to contact study authors" It may be helpful to add a deadline here, for example, to say no more attempts at contact will be made after two months.

We have now added information regarding a follow-up deadline, as per your suggestion above. We have added 'with no more attempts at contact made after two months' to the relevant sentence (page 11).

**- "It will also explore relationships in the data, the development of a theoretical framework (if relevant), and assess the strength of the evidence for the conclusions drawn from the synthesis, as per formal guidelines on the conduct of a narrative synthesis"
The sentence above promises a lot of work being done, without providing any detail. Please explain how these tasks will be done. For example, I would not consider no.17 in the PRISMA checklist completed. You may also elaborate on the way the 'descriptive overview' will be structured, for example by outcome or age group or geographic region.**

Thank you for your feedback regarding the above, we can see that some further details may be helpful for the reader. We have now inserted the following text below, in the interests of greater clarity (page 12):

'If meta-analyses are not possible due to insufficient data on the rates or associated characteristics of self-harm, or if there is substantial heterogeneity, a narrative synthesis will be created (as per established guidelines on the conduct of a narrative synthesis)⁵². This descriptive overview will provide information in both text and table format to summarise and explain the included study findings, based on outcome, associated characteristics of self-harm, and geographic region. From this, we may also develop a theory⁵² of self-harm among unaccompanied asylum seekers and refugee minors, and the factors that influence it, and an assessment of the overall robustness of this summary.'

In regard to your last point above, our 'descriptive overview' will be structured according to study year(s), design, size, and location of the study sample, measure(s) used to report self-harm (page 12).

Reviewer: 2

**Dr. Darren Courtney, Centre for Addiction and Mental Health
Comments to the Author:**

The manuscript is a protocol for a systematic review of prevalence rates. The population are minors (under 18 years old) who are seeking asylum or have refugee status. The outcome variable is self-harm. Exposure/comparator groups and timing of outcomes are not specified.

The authors anticipate that prevalence rates will be relatively high and propose that this information could help guide investment in treatment.

General:

The manuscript is, for the most part, well written, clear and concise. PRISMA-P reporting guidelines are followed. Results of the review may be broadly applicable as samples of interest may be found across the globe.

Many thanks for your helpful feedback, we are pleased to hear that you think our manuscript is well written, clear, and concise.

Some revisions could strengthen the manuscript.

(1) Reference to an established method of systematic reviews is needed. One option is to consider reference to the JBI Evidence Synthesis Manual from the Joanna Briggs Institute. Munn Z, Moola S, Lisy K, Riitano D, Tufanaru C. Chapter 5: Systematic reviews of prevalence and incidence. In: Aromataris E, Munn Z (Editors). JBI Manual for Evidence Synthesis. JBI, 2020. Available from <https://synthesismanual.jbi.global>. <https://doi.org/10.46658/JBIMES-20-06>

Specifically, this manual guides investigators on:

(a) determination of the need for a review (Section 1.2, relevant for page 7, line 3 of manuscript)

Thank you for this recommendation, based on the JBI evidence synthesis manual. Preliminary searches in PROSPERO, Pubmed/MEDLINE, and the DARE database were conducted prior to registering our protocol with PROSPERO, though we appreciate that we did not specifically highlight this for the reader. We have now clearly outlined this information in the relevant section mentioned above, in line with the JBI evidence synthesis manual. We have also added that searches were conducted in the JBI's own online journal, JBI Evidence Synthesis. The text (page 6) now reads:

'Earlier systematic reviews have largely focused on the mental health and wellbeing of unaccompanied minors,^{19 24-26} or the mental health of refugee minors.²⁷ However, preliminary searches conducted by the authors for the purposes of this study (in PROSPERO, the PUBMED/MEDLINE and DARE databases, and the JBI Evidence Synthesis journal)²⁸ indicate that no reviews have examined self-harm (elsewhere also described as non-suicidal self-injury [NSSI], and defined as the deliberate, self-inflicted destruction of body tissue, without suicidal intent, and for purposes not socially or culturally sanctioned²⁹) in unaccompanied asylum seekers and refugee minors globally.'

(b) need to conduct review steps (i.e, title/abstract screening, full-text selection, data extraction, quality assessment), either showing adequate inter-rater reliability/agreement prior to a single rater proceeding with the step; or conducting a step in duplicate (section 5, relevant for methods section). Having a second rater checking the work of another rater is too vulnerable to bias.

In line with the JBI evidence synthesis manual, and in accordance with your recommendations, as well as those from reviewer 1 (see above), we have now inserted further clarifying details for the reader regarding the review steps taken (pages 10-11):

'All titles and abstracts will be screened for inclusion by the primary author (KH), with 20% screened by the second author (RB). After 20% of the papers identified in the search strategy have been double screened, we will reassess our eligibility criteria to ensure that they are relevant to the studies that are identified. The reassessment process will involve a discussion between the reviewers, with any differences or uncertainty regarding study inclusion resolved by a third researcher. The overall inter-rater reliability for the title and abstract screening will be calculated using Cohen's kappa statistic.⁴⁷ Studies will be coded as either 0=Does not meet eligibility criteria, or 1=Meets eligibility criteria or the full-text article needs to be screened to confirm eligibility. After 20% of the citations have been screened, the eligibility criteria will be reassessed through discussion between all researchers to ensure that they are relevant to the studies that have been identified. If applicable, the updated eligibility criteria will be used for the remaining screening.

After title and abstract screening is complete, all remaining full-text articles will be independently screened by KH and RB, with any conflicts related to study inclusion resolved through discussion with a third researcher. Where clarification is needed to determine eligibility, we will make a maximum of three attempts to contact the original study author(s), with no more attempts at contact made after two months.'

(2) Reference to the inconsistent literature on self-harm nomenclature would also be helpful. For example,

- O'Carroll, P. W., Berman, A. L., Maris, R. W., Moscicki, E. K., Tanney, B. L., & Silverman, M. M. (1996). Beyond the Tower of Babel: a nomenclature for suicidology. *Suicide and Life-Threatening Behavior*, 26(3), 237-252.
- Silverman, M. M., Berman, A. L., Sanddal, N. D., O'carroll, P. W., & Joiner Jr, T. E. (2007a).

Rebuilding the tower of Babel: a revised nomenclature for the study of suicide and suicidal

behaviors. Part 1: Background, rationale, and methodology. *Suicide and Life-Threatening Behavior*, 37(3), 248-263.

- Silverman, M. M., Berman, A. L., Sanddal, N. D., O'carroll, P. W., & Joiner Jr, T. E. (2007b). Rebuilding the tower of Babel: a revised nomenclature for the study of suicide and suicidal behaviors. Part 2: Suicide-related ideations, communications, and behaviors. *Suicide and Life-Threatening Behavior*, 37(3), 264-277.
- De Leo, D., Goodfellow, B., Silverman, M., Berman, A., Mann, J., Arensman, E., ... & Kolves, K. (2021). International study of definitions of English-language terms for suicidal behaviours: a survey exploring preferred terminology. *BMJ open*, 11(2), e043409.

Thank you, in line with your suggestions, we have now inserted 2 further references, as well as a brief comment for the readers regarding the long-standing inconsistent self-harm nomenclature (page 8):

'Indeed, definitions of self-harming behaviour have varied historically,^{40 41} and behavioural intent is difficult to measure. Despite this, as self-harm and suicide attempts are motivationally and qualitatively distinct from one another,⁴² differentiating between the two – *where possible* – is critical to inform the provision of appropriate physical and mental health care.'

Hawton and colleagues' definition of self-harm does include suicide attempts; several times in the manuscript authors suggest that self-harm is mutually exclusive from suicidal behaviour – when this is not the case in Hawton's definition; this needs further clarification.

Thank you for your suggestion, we've now altered the definition we are using. On pages 6 and 9, the text now reads: For the purposes of this review, self-harm, elsewhere also described as non-suicidal self-injury [NSSI], is defined as the deliberate, self-inflicted destruction of body tissue, without suicidal intent, and for purposes not socially or culturally sanctioned.²⁹ On page 9, we have also added that 'it may include behaviours such as cutting, burning, biting, and scratching skin.'²⁹

(3) The term “unaccompanied asylum seekers or refugee minors” is confusing. Namely, all of the adjectives refer to minors. I wonder if “unaccompanied minors who are refugees or seeking asylum” might be a better fit; an abbreviated version (e.g, UMRSA) could be used once defined.

Thank you for your comments regarding potential terms for the two populations we are investigating. The standard and most common abbreviation for unaccompanied refugee minors in the literature, as well as in practice, is URM. In our study, we are investigating both unaccompanied asylum seekers and refugee minors – two distinct groups. As a consequence, we would prefer to list our two populations in full, rather than invent a new term (for example, UMRSA) that may be confusing for readers who are most familiar with the standard URM abbreviation.

I also recommend that the operational definitions of these terms can be moved in the earlier part of the introduction as it contextualizes the literature to date described.

Thank you for your suggestions regarding the positioning of the operational definitions of the above terms.

In accordance with your recommendation, we have now moved the definitions of refugees and asylum seekers from page 8 to page 4 – the very first paragraph of the introduction section. The definition of self-harm has also been moved to page 6, as well as being mentioned in the ‘Outcomes’ section on page 9.

(4) The potential impact of the results of the current study should be made clearer. If a high prevalence is found, is there a way to determine if it is higher than the general population <18 years old (which is already high)? Is the goal to establish whether refugee status/asylum-seeking is an independent risk factor for self-harm? How could the results specifically inform treatment efforts? Why is self-harm an important outcome to target in treatment, particularly if not suicidal?

Thank you for your feedback. Where possible, we will attempt to determine whether the prevalence of self-harm in our population/s of interest is higher than the general population. We will be including (and synthesizing evidence from) comparison studies in our review, where possible, for that reason. We have inserted the following sentence (page 9) from our PROSPERO registration, for greater clarity for the reader:

‘Comparison studies involving accompanied asylum seekers and/or refugee minors, as well and the general/native population in countries of origin, transit, and/or resettlement will also be included, where available.’

On page 9, we have also added some further references to self-harm in young people and linked these to the literature we had previously highlighted regarding the elevated self-harm risk among young asylum seekers and refugees. Specifically, we have highlighted that ‘As self-harm is common among young people,^{30 31} and this risk is further elevated among younger asylum seekers and refugees and those with experiences of trauma,^{32 33} it is conceivable that unaccompanied asylum seekers and refugee minors are at even greater risk of self-harm.’ As stated above, comparison studies with the general population will also be included, where available.

In regard to how the results might specifically inform treatment efforts, we have now inserted the following 2 clarifying sentences (as well as two additional relevant references) to the text on page 9:

‘Should a higher prevalence of self-harm among unaccompanied asylum seekers and refugee minors be identified, these findings would elucidate the need for providing mental health support at the earliest available opportunity for such minors. Furthermore, as the social, familial, and public health costs of self-harm are known to be high,^{34 35} the synthesis of such evidence may help to inform distribution of scarce public resources.’

In response to your last comment above, as individuals can die (accidentally) from self-harm at any time, self-harm is a critical outcome to target in treatment, including for individuals who may not be

suicidal. In order to highlight the importance of this for the reader, we have also inserted the following text on page 9:

'Indeed, given that every act of self-harm (including explicitly non-suicidal actions and behaviour) has the potential to be lethal as a result of accident or misfortune, self-harm (irrespective of intent) is an extremely important outcome to target.'

(5) Prior literature should be further clarified. When a study is referred to, the study design and sample size and age ranges need to be clearly articulated to justify the need for the current study. If proportions are listed, indicate 95% CI's or p-values to say how the samples are different.

All details from prior literature regarding study design, sample size, and age ranges, as well 95% CI's or p-values – where available – have now been added (page 7).

(6) Further detailed recommendations:

(a) In the abstract, the inclusion/exclusion criteria need to match the methods section. In the abstract it seems very broad, in the methods section it seems very narrow: “cohort studies”.

In keeping with the 300-word limit for the abstract, we have added as much additional information regarding the inclusion/exclusion criteria from the methods section as journal rules permit. The inclusion/exclusion criteria in the abstract now read:

'We will include all types of study design (except single case studies) that examine the prevalence of self-harm in unaccompanied asylum seekers and/or refugee minors. We will exclude dissertations, conference abstracts, letters, book chapters, editorials, study registrations, clinical trials, case-control studies, registered protocols, and qualitative studies. Only studies reporting on participants aged <18 years will be eligible for inclusion.'

(b) EU needs to be spelled out on the first instance.

This abbreviation has now been amended (page 4).

(c) Pg 6, In 36: When it's a group of authors being cited, add “and colleagues...”

Thank you, we are aware of this convention. As three groups of authors needed to be listed in short succession in one sentence, we made the decision to refer to two of the authorship groups in a more expedient fashion. In response to your feedback, however, we have amended the sentence so that it now reads: 'a Swedish study by Ramel and colleagues¹⁹ was included in both von Werthern and colleagues³⁶ and Garguilo and colleagues reviews³⁸'.

(d) Pg 6, In 57: “Neither” □ “None”

This has now been altered so it reads 'None' (page 7).

(e) Inclusion/exclusion: What about case-control studies? Qualitative studies? Grey literature? Published protocols? Clinical trials?

Thank you for your feedback. To our statement regarding exclusions in the 'study design' section, we have now added that single case studies, dissertations, conference abstracts, letters, book chapters, editorials, study registrations, clinical trials, case-control studies, registered protocols, and qualitative studies will be excluded (page 9).

Our grey literature search is highlighted in the 'Information sources and search strategy' section, in the paragraph below the 'study design' section (page 10).

For extra clarity for the reader, however, and in line with your recommendation, we have now added the sentence 'we will also include grey literature (see "Information sources and search strategy" below)' to page 9.

(f) Specific search strategy can go in an Appendix/Supplementary material, with a brief general description in the main text.

The full search strategy has now been included as a Supplemental appendix 1 (see page 10), with a brief general description in the text, in line (also) the editor's feedback (above).

(g) Having a priori criteria to progress to a meta-analysis would be ideal (e.g., number of papers, consistency across papers in design/samples/outcome measures, etc.).

We have indicated (page 11) that 'if a sufficient number of studies report on the rates of self-harm, we will conduct meta-analyses to calculate pooled estimates of self-harm rates.' In response to your feedback, we have now also inserted the phrase 'and homogeneity between studies...' in the above sentence.

VERSION 2 – REVIEW

REVIEWER	Courtney, Darren Centre for Addiction and Mental Health, Psychiatry
REVIEW RETURNED	13-Mar-2023

GENERAL COMMENTS	Thank you for making changes in response to my suggestions. There are a few outstanding things to consider prior to publication: (1) Abstract: The inclusion criteria and exclusion criteria now contradict each other – “we will include all types of study design..... We will exclude [list of many types of study design]”. Consider revising to succinctly capture what is in the main text. (2) Page 6, line 38: “As self-harm is common among young people [30, 31], and this risk is further elevated among younger asylum seekers and refugees and those with experiences of trauma [32, 33], it is conceivable that unaccompanied asylum seekers and refugee minors are at even greater risk of self-harm.” The above sentence needs more clear phrasing; the second part “and this risk is further elevated among younger asylum seekers and refugees ...” would suggest your research question has already been answered. Can you more clearly distinguish the population you are referring to in this sentence and “unaccompanied asylum seekers and refugee minors”? (3) The rationale for self-harm as an outcome of interest needs further justification. It is quite rare for non-suicidal injury to lead to death or even needing medical attention. It is, however, a risk marker for severe mental illness, risk marker for suicide, and risk marker for all-cause mortality (e.g., https://doi.org/10.1016/S2352-4642(19)30373-6); these are more convincing reasons to draw attention to this outcome. (4) Page 11, line 10: Consider random selection of papers to be rated by second rater – reduces risk of bias.
---

	(5) Page 11, line 22: I would anticipate that the kappa statistic might inform next steps – how will you proceed if it is high (e.g. proceed with single ratings for remaining titles/abstracts)? What about low (proceed with duplicate ratings for remaining titles/abstracts) ? What cut-off will you use to make the decision? (6) Page 12, line 23: JBI Section 5.4.5 and 5.5.7 indicates that critical appraisal (i.e. ROB assessments) and data extraction should be done in duplicate independently – with later discussion of inconsistencies; not done by one reviewer and then checked by another. (7) Page 13, line 27: “we may also develop a theory”. Ideally, the background section would include existing, supported theories of why people self-harm and why you think that the population of interest might be more susceptible than the general population. I would not think you would develop a theory “from scratch” based on this review – maybe build on prior theories? A few minor things: (1) Page 4, line 36; “seeker” □ “seekers” (2) Page 11, line 33: “independently screened by KH and RB” □ “independently screened in duplicate by KH and RB”
--	--

VERSION 2 – AUTHOR RESPONSE

Reviewer: 2

**Dr. Darren Courtney, Centre for Addiction and Mental Health Comments to the Author:
BMJ-Open 2022-069237.R1**

Reviewer comments:

Thank you for making changes in response to my suggestions.

Thank you for your time and feedback once again.

There are a few outstanding things to consider prior to publication:

(1) Abstract: The inclusion criteria and exclusion criteria now contradict each other – “we will include all types of study design.... We will exclude [list of many types of study design]”. Consider revising to succinctly capture what is in the main text.

We have now inserted the italicized text below (on page 2) for greater clarity:

‘With the exception of single case studies, clinical trials, and case-control studies, we will include all types of study design that examine the prevalence of self-harm in unaccompanied asylum seekers and/or refugee minors. We will exclude dissertations, conference abstracts, letters, book chapters, editorials, study registrations, registered protocols, and qualitative studies.’

(2) Page 6, line 38: “As self-harm is common among young people [30, 31], and this risk is further elevated among younger asylum seekers and refugees and those with experiences of trauma [32, 33], it is conceivable that unaccompanied asylum seekers and refugee minors are at even greater risk of self-harm.”

The above sentence needs more clear phrasing; the second part “and this risk is further elevated among younger asylum seekers and refugees ...” would suggest your research question has already been answered. Can you more clearly distinguish the population you are referring to in this sentence and “unaccompanied asylum seekers and refugee minors”?

We have amended the sentence slightly so that it now reads:

‘As self-harm is common among young people [30, 31], and this risk is further elevated among those with experiences of trauma [32], it is conceivable that unaccompanied asylum seekers and refugee minors are at even greater risk of self-harm due to their younger age and prior experience of a range of potentially traumatic events in the pre-, peri-, and post-migration phases [10].’

(3) The rationale for self-harm as an outcome of interest needs further justification. It is quite rare for non-suicidal injury to lead to death or even needing medical attention. It is, however, a risk marker for severe mental illness, risk marker for suicide, and risk marker for all-cause mortality (e.g., [https://doi.org/10.1016/S2352-4642\(19\)30373-6](https://doi.org/10.1016/S2352-4642(19)30373-6)); these are more convincing reasons to draw attention to this outcome.

We thank the reviewer for their suggestions for additional justifications here – in crafting our manuscript, we’ve attempted to be mindful of the journal word limit for protocol papers.

Please find the small additions made to relevant text (page 6) in italics below:

‘Indeed, *given that self-harm is a risk factor for severe mental illness, as well as for all-cause mortality* [33], self-harm is an outcome of marked importance. Furthermore, as every act of self-harm (including explicitly non-suicidal actions and behaviour) has the potential to be lethal as a result of accident or misadventure, and *self-harm is the strongest risk factor for suicide* [33, 34], self-harm (irrespective of intent) is an extremely important outcome to target.’

(4) Page 11, line 10: Consider random selection of papers to be rated by second rater – reduces risk of bias.

We have now added the word ‘random’ to the relevant text on page 11, so that it is clear that a random 20% will be screened by the second author (RB) (page 11).

(5) Page 11, line 22: I would anticipate that the kappa statistic might inform next steps – how will you proceed if it is high (e.g. proceed with single ratings for remaining titles/abstracts)? What about low (proceed with duplicate ratings for remaining titles/abstracts)? What cut-off will you use to make the decision?

In response to reviewer suggestion, we have now inserted the following text into the relevant paragraph on page 11:

‘If reliability is low (<0.40),⁴⁷ the authors will review the eligibility criteria, double-code a second random 10% of retrieved studies and recalculate Cohen’s kappa statistic.’

(6) Page 12, line 23: JBI Section 5.4.5 and 5.5.7 indicates that critical appraisal (i.e. ROB assessments) and data extraction should be done in duplicate independently – with later discussion of inconsistencies; not done by one reviewer and then checked by another.

We’ve now altered the text, in line with reviewer suggestions – see italicized text below for changes made (page 12):

‘The quality and risk of bias will be *independently appraised in duplicate* by KH and RB. *The two reviewers will then compare scores*, with any uncertainty resolved through discussion and consensus before allocation of final appraisal scores.’

(7) Page 13, line 27: “we may also develop a theory”. Ideally, the background section would include existing, supported theories of why people self-harm and why you think that the population of interest might be more susceptible than the general population. I would not

think you would develop a theory “from scratch” based on this review – maybe build on prior theories?

We thank the reviewer for their thoughts in relation to this point. Both authors are members of the newly established (i.e., in 2022) international network of transcultural self-harm and suicidology, and one of our first planned outcomes is an editorial addressing the lack of theory in this space. At the time of writing, only one paper has been published and theory is, by definition, extremely underdeveloped. As such, at this point in time, we do not wish to borrow fully from theories related to other populations, which may not be wholly applicable to this population.

A few minor things:

(1) Page 4, line 36; “seeker” → “seekers”

This typo has now been corrected.

(2) Page 11, line 33: “independently screened by KH and RB” → “independently screened in duplicate by KH and R

We have now inserted the word ‘duplicate’ in the relevant sentence on page 11.